# Resolution of Resilience: Empirical Findings on the Challenges Faced and the Mitigation Strategies Adopted by Community Health Workers (CHWs) to Provide Maternal and Child Health (MCH) Services during the COVID-19 Pandemic in the Context of Odisha, India

**DOI:** 10.3390/healthcare10010088

**Published:** 2022-01-03

**Authors:** Bijaya Kumar Mishra, Srikanta Kanungo, Kripalini Patel, Swagatika Swain, Subhralaxmi Dwivedy, Subhashree Panda, Sonam Karna, Dinesh Bhuyan, Meena Som, Brajesh Raj Merta, Debdutta Bhattacharya, Jaya Singh Kshatri, Subrata Kumar Palo, Sanghamitra Pati

**Affiliations:** 1ICMR-Regional Medical Research Centre, Bhubaneswar 751023, India; bijayak.mishra@icmr.gov.in (B.K.M.); srikanta.kanungo@icmr.gov.in (S.K.); kripalinie.patel@gmail.com (K.P.); silaswagatika@gmail.com (S.S.); subhralaxmid@gmail.com (S.D.); subhashree.panda98@gmail.com (S.P.); karna.sonam110@gmail.com (S.K.); dinesh_bhuyan@yahoo.com (D.B.); drdebdutta.bhattacharya@yahoo.co.in (D.B.); jsk.icmr@outlook.com (J.S.K.); 2United Nation Children’s Fund, Surya Nagar, Bhubaneswar 751003, India; msom@unicef.org (M.S.); bmerta@unicef.org (B.R.M.)

**Keywords:** COVID-19, community health worker, maternal and child health, odisha, pandemic

## Abstract

Community health workers (CHW) faced increased challenges in delivering maternal and child health services during the current COVID-19 pandemic. In addition to routine services, they were also engaged in pandemic management. In view of a dearth of evidence, the current study explores the challenges faced by CHWs while rendering maternal and child health services. A qualitative study through in-depth interviews (IDI) and focus group discussions (FGD) in six districts of Odisha was conducted from February to April 2021. Data were analyzed using MAXQDA software. Personal-level challenges, like lack of family support, stress, and fear of contracting COVID-19; facility-level challenges, like transportation problems and inadequate personal protective measures; and community-level challenges, like stigma, resistance, and lack of community support were major hindrances in provisioning routine MCH services. Prevailing myths and misconceptions concerning COVID-19 were factors behind stigma and resistance. Sharing experiences with family, practicing yoga and pranayam, engaging ambulance bikes, financial assistance to mothers, counseling people, and involving community leaders were some effective strategies to address these challenges. Development and implementation of appropriate strategy guidelines for addressing the challenges of frontline warriors will improve their work performance and achieve uninterrupted MCH services during pandemics or similar health emergencies.

## 1. Introduction

The COVID-19 pandemic has demonstrated the need for investment in the health workforce [1]. Ebola virus and human immunodeficiency virus (HIV) illnesses have taught us the importance of community involvement in decision making, as well as the formulation of cheap and effective strategies for disease prevention [2]. Similarly, the COVID-19 pandemic has provided a unique scope to understand the working conditions of community health workers (CHWs) involved in the provision of healthcare services including maternal and child health services for community people. CHWs such as Accredited Social Health Activists (ASHAs) and Auxiliary Nurse Midwives (ANMs) provide routine healthcare services under various health programs. While the major role of ASHAs is to mobilize beneficiaries for availing the health services, ANMs are primarily responsible for the provision of health services either at sub-center (SC) facilities or through outreach activities. Lady health visitors (LHVs) mainly supervise community-based activities and provide mentoring support to frontline workers (ANMs and ASHAs). Though anganwadi workers (AWWs) are also frontline workers, they were not considered in our study, owing to their nature of work and the different department they belong to. During the pandemic, community health workers (CHWs) have undertaken a series of new tasks in addition to their routine services, including maternal and child health (MCH) services. These new tasks include conducting household surveys for COVID-19 case detection, creating awareness about COVID-19, COVID testing, door-to-door visits, and contact tracing. Along with these additional tasks, they have also contributed to their routine job responsibilities in community [3,4]. Researchers have identified the shortfalls in the provision and utilization of routine MCH services during the current pandemic. A drop in utilization of antenatal care (ANC) and post-natal care (PNC) [5,6], institutional delivery [7], routine immunization services [8,9], and family planning services [10] signifies the lapses in MCH care due to the COVID-19 pandemic. Prior to the onset of the pandemic, the state of Odisha was making great efforts to improve its maternal and child health indicators [11] to achieve the figures targeted under sustainable development goals (SDG).

Various research studies have highlighted the challenges faced by CHWs while performing their multiple roles and responsibilities in different parts of India, even prior to this pandemic [12,13,14]. Therefore, when their workload increased to accommodate pandemic-related responsibilities during this unprecedented time, it is likely that their challenges must have increased manifold [11]. There is a dearth of evidence regarding the challenges faced by CHWs while delivering healthcare services, particularly MCH services, during the COVID-19 pandemic, specifically in Eastern India. The current study was intended to delve into the challenges encountered by the CHWs while delivering MCH services during the COVID-19 pandemic and to gain an understanding of the different strategies adopted to overcome those challenges.

## 2. Methodology

### 2.1. Study Design, Setting, and Data-Collection Procedure

We carried out a qualitative study among CHWs (ASHA, ANM, LHV) using in-depth interviews (IDI) and focus group discussions (FGD). The study was conducted in six of thirty districts in Odisha state (India) from February to April 2021. A multistage sampling method was adopted for selecting the districts, blocks, and health facilities. The state has three revenue divisions (Northern, Central and Southern). From each division, two districts were randomly selected for better representativeness. Two administrative blocks under each district were again randomly selected. From each of the study blocks, one community health center (CHC), one primary health center (PHC), and one sub-center (SC) were selected to identify and recruit the study participants. While SCs are the lowest level of healthcare facilities, they are monitored by the PHC they belong to; and similarly, the PHCs are monitored by the CHC they belong to. The study site characteristics and details about the IDIs and FGDs among different CHW participants are presented in Figure 1 and Table 1.

### 2.2. Data Collection and Analysis

The participants were interviewed in Odia (the vernacular language of Odisha), using predesigned and pretested IDI and FGD interview guides. Both guides consisted of questions, along with probes and prompts, pertaining to eliciting information regarding the nature of participants’ work, their situation during the COVID-19 pandemic, support and challenges from the health system, and community response. The guides were further reviewed by subject experts and pilot tested, and necessary changes were made before doing the IDIs and FGDs among study participants. For FGDs, participants (ASHA or ANM) from the study area were invited to the concern block CHC at a scheduled time. Prior to each interview, the research objectives and voluntary essence of the study were explained to the participants, and informed written consent was obtained. All interviews were carried out by a team of qualified and trained researchers and audio-recorded with the participants’ consent. While conducting all the FGDs and IDIs, appropriate precautions and procedures were adhered to, as per the prescribed norms, to prevent the spread of COVID-19. Researchers with sound knowledge in both Odia and English language transcribed the audio recordings in Odia and later translated those into English. The translations were cross-verified by the project investigators to avoid any bias in interpretation. We followed the source-and-investigator triangulation approach to enhance the credibility of study findings. A qualitative content analysis method following the conventional approach [15] was adopted to analyze the collected data. MAXQDA software was used for analysis of the transcripts and generation of codes. Three researchers reviewed the code tree and came up with the themes and categories by consensus. Categories were organized, and themes and subthemes were generated. The study findings were summarized and narrated accordingly. The developed code tree is given in Appendix A.

### 2.3. Ethical Consideration

Ethical approval for this research was obtained from the Institutional Ethics Committee of ICMR-Regional Medical Research Centre, Bhubaneswar and the State Research and Ethics Committee, Department of Health and Family Welfare, Govt. of Odisha.

## 3. Result

The study found various contextual factors that hindered CHWs in delivering maternal and child health (MCH) services during the COVID-19 pandemic. Our study result encompasses two overarching themes: 1. Challenges encountered in delivering MCH services; and 2. Different strategies adopted to overcome those challenges. Further, these challenges and strategies were categorized at three different levels (personal, community, and facility). While all the participants more or less described their challenges, there were differences in opinion according to category of participants, geography of study area, and type of interview (FGD or IDI). However, for a better understanding, the findings are collated and presented systematically.

### 3.1. Theme 1: Challenges Encountered in Delivering MCH Services

#### 3.1.1. Personal-Level Challenges

Focusing on pandemic-related activities and continuing to provide essential services were already enormous tasks for CHWs. During such an adverse situation, lack of family support was one of the most significant challenges faced by a few participants.
*“Every day, my family would ask me to leave my job as there was risk of contracting the infection. They told me that I would be responsible if they die.”**(ANM)*

The workload of the CHWs increased. In addition to their regular duties and responsibilities, including MCH services, they were assigned COVID-related duties. For instance, they were engaged in COVID-19 testing (rapid antigen test), contact tracing, follow-up of quarantined patients, and daily reporting, etc.
*“We were involved in so many duties. It was a test of patience for us too.”**(ANM)*

The challenges in delivering MCH services increased with increased inflow of migrant people in local areas. Delivering MCH services to these migrant beneficiaries was also a major task assigned to CHWs.
*“Madam, the issue emerged when the migrant beneficiaries increased, and we had difficulty in screening these pregnant women. Women with 8–9 months of pregnancy were sent to the nearest delivery points, where RRT teams visited them on a priority basis. After their test results only, they were taken for obstetrical checkups.”**(ANM)*

Most CHWs had the fear of contracting the infection and spreading it in their family. However, despite their fear, they performed their duties.
*“Yes, we had the fear of getting the infection because every day we visited the community and the hospital. So, I thought I might get the infection. But by the grace of God, I did not get the infection.”**(ASHA)*

CHWs showed a remarkable resilience and professional dedication despite the fear of becoming infected and infecting others. Participants stated that they had strong willpower to perform their work, even during the pandemic.
*“Actually, I did not have much fear; the only important thing was to take appropriate measures like using sanitizer, not eating outside food, using mask, washing vegetables properly, avoid gathering etc. I adhered to all the measures.”**(ANM)*
*“No madam, I have a strong will power and I motivate myself daily for any type of work in any situation.”**(LHV)*

#### 3.1.2. Facility-Level Challenges

One of the major activities of ASHAs is to accompany expecting mothers to health centers as a “birth companion” for their delivery. During the nationwide lockdown, many ASHAs faced transportation difficulties returning home after settling the patient in the hospital because no public transport system was available during that period. A few ASHAs also mentioned out-of-pocket expenditures incurred for transportation.
*“We faced a lot of problems. The ambulance took the patient to hospital, but that vehicle didn’t drop us back at our home.”**(ASHA)*

In view of COVID restrictions, certain health facilities did not allow the ASHAs to enter the facility with the pregnant mother.
*“It is okay if they don’t allow us into the hospital but… they should allow us until the patient gets registered in the hospital.”**(ASHA)*

Though most CHWs had access to necessary personal protective equipment (PPE), some raised concerns of shortages due to irregular supply during the initial phase of the pandemic.

#### 3.1.3. Community-Level Challenges

*Stigma of infection:* Participants narrated various stigmas among community members while visiting them to provide routine and COVID-19-related health services. In some communities, reluctance to accept health services, owing to perceptions that CHWs might be infected, was a significant cause of lack of community support for some CHWs.
*“Most people did not allow us to enter their home. They told us that we work in the hospital and hence we must have been infected. We had to work hard to make them understand that our working in the hospital didn’t mean we were COVID positive.”**(ANM)*

Despite their long relationships and services during the pandemic, some CHWs faced stigma from community members, which hampered delivery of MCH and other health services.
*“We did not enter anyone’s house. We used sticks to knock on their doors.”**(ASHA)*
*“We would take all the logistics used for village health and nutrition day (VHND) like BP instruments, thermometer etc.to their house for check-up. Still they were a little hesitant because in COVID situation nobody was allowed to visit their house. Some people directly denied and closed their door.”**(ANM)*

*Resistance from Community:* The participants also mentioned a few incidences of resistance from the community. There were instances of boycotting households with a positive COVID-19 case. This was an important reason why people were not interested in getting tested. CHWs were assigned the responsibility of counseling migrants and testing them for COVID. While doing so, many CHWs had to face resistance from some migrants and even had to tolerate foul language in such instances.
*“Community people (migrants) scolded us; we should not discuss that now.”**(ANM)*

### 3.2. Theme 2: Strategies Adopted to Overcome the Challenges

#### 3.2.1. Personal-Level Strategies

In order to ensure MCH services during the COVID-19 pandemic, participants had to adopt some of their own strategies, other than adhering to the government guidelines. They stated that though there were some instances of stigma and resistance from the community, they continued to deliver their services conducting door-to-door visits.
*“Counseling was the only effective way to do our duty during the COVID times. People did not recognize it initially, but after several attempts, they understood the scenario.”**(ANM)*

Participants also revealed that they felt relaxed after sharing their daily experiences with family members. A few CHWs also stated that they did yoga and meditation to overcome their work-related stress and anxiety.

#### 3.2.2. Facility-Level Strategies

To address transportation-related issues, CHW shad to use existing local resources, such as “bike ambulances” in some districts. Some participants also mentioned that financial assistance was provided to pregnant mothers from the health system to avail transportation services.
*“Government provides 500 rupees for transportation. If ambulance did not arrive, we had to hire auto or car for transportation to health center.”**(ASHA)*

CHWs also recommended various strategies that could be adopted to help them carry out their roles and responsibilities more effectively. They suggested taking measures to safeguard them during work, provision of transportation facilities, adequate training, and health insurance.

#### 3.2.3. Community-Level Strategies

CHWs adopted strategies to involve the community and generate awareness of COVID-19 among them. For this, they conducted door-to-door visits and also did wall paintings, poster displays, and used loudspeakers at the community level.
*“We helped each other and worked a lot. Wall paintings, creating awareness through loudspeakers, posters—everything was done to create awareness in the community. Survey was conducted every day. We also provided our contact numbers to the community members. This awareness was effective.”**(ANM)*

Many community leaders took proactive initiative to reduce the stigma and resistance in the community and to help CHWs do their duty.
*“During our door-to-door awareness drive, the Sarapanch (chief of the village Panchayat) aided us. The Sarapanch came to our rescue when we were targeted by some people and faced stigma. He convened meetings with small groups and tried to persuade people that we were here only for their benefit.”**(ANM)*

A conceptual framework based on the challenges faced, their effects, and strategies adopted by CHWs is depicted in Figure 2. While the challenges and strategies are based on the comments of the participants, the effects resulting from the challenges are as perceived by the participants, reflected in their verbatim interviews.

## 4. Discussion

### 4.1. Challenges Faced by CHWs

To our knowledge, this is the first study in eastern India to explore the challenges faced by CHWs and the strategies they adopted to ensure MCH services during the COVID-19 pandemic in Odisha. During the pandemic, CHWs were at the forefront of mitigation of the health challenges at the community level [16]. Personal-level challenges, like lack of family support, stress, and fear of contracting the infection; facility-level challenges, like logistic problems relating to transportation and inadequate personal protective equipment during the initial phase; and community-level challenges, like stigma, resistance and lack of community support were important hindrances to the provision of MCH services during the COVID-19 pandemic. A potential reason behind the stigma and resistance in some communities is due to myths and misconceptions surroundingCOVID-19. Similar to our study findings, other studies have revealed that CHWs encountered threats and hostility in their field areas [17,18].

### 4.2. Effects of Challenges

Our study also reveals the effects of such challenges, as perceived by CHWs. Participants mentioned that lack of family support, managing the migrant beneficiaries, and fear of contracting COVID-19 infection in the event of increased workload during the pandemic led to fear, stress, and anxiety. A study by Gupta et al. highlighted mental health problems, such as anxiety, depression, and stress-related disorders, among CHWs during this pandemic [19]. Inadequate availability of PPE measures (during the initial period of the pandemic) further added to their fear, safety concerns, and anxiety [19]. ASHA workers who accompanied expecting mothers to health facilities for delivery were denied entry and stay with the mother, and also faced problems returning back to their homes, resulting in their reduced trust in the system. Myths and misinformation about COVID-19 due to lack of awareness resulted in stigma, which led to challenges in the form of low community support and reluctance to accept healthcare services. Instances of community resistance affected CHWs’ morale and motivation in performing their assigned work [3].

### 4.3. Strategies Adopted

In order to overcome these challenges, CHWs adopted different strategies amidst the COVID-19 pandemic. Sharing work experiences with family members and practicing regular yoga and pranayama were stress busters to address personal-level challenges. A study conducted by Htay et al. [20] revealed that “getting family support” and “positive thinking” were coping methods among healthcare workers during the COVID-19 pandemic. In India, in order to manage the stress and anxiety among CHWs, training modules under the title “Promoting Health Worker Safety; A Priority for Patient Safety during COVID -19 Pandemic and Beyond” was promoted by the Ministry of Health and Family Welfare (MoHFW). Furthermore, to address transportation-related challenges, ambulance bikes were engaged by the health system, and even extra financial assistance was extended to expecting mothers for transportation. Although during the initial period of the pandemic, there was some scarcity of PPE, the issue was sorted out through prompt initiative in procurement and supply-chain management by the state health system. Some research studies [21,22,23] have explored the strategies adopted in India to enhance the accessibility of protective gear in healthcare settings.

Prevailing myths and misinformation concerning COVID-19, along with a low level of health awareness, were the reasons responsible for the stigma and resistance of community members, which minimized their support to and cooperation with CHWs. To address this challenge, CHWs adopted strategies such as improving community awareness through information communication using counseling, wall paintings, poster campaigns, and loudspeakers. Additionally, involving community leaders, representatives, and influential persons was effective in addressing these challenges. Some CHWs, with the help of elected community representatives, organized meetings in different communities to dispel myths around COVID-19. This helped to address the stigma and resistance of community members. The government of India provided strategies to train ASHA workers through Panchayati Raj institutions for early identification of COVID-19 cases [24]. Additionally, the participants also suggested initiatives such as provision of security in work environment, provision of transportation facilities, training on COVID-19, and health insurance for better performance. Establishing aggressive management teams at the community level could be an effective measure to overcome the community resistance.

### 4.4. Implications for Policy and Practices

The findings from our study provide evidence concerning the challenges encountered by the CHWs while rendering routine services, including MCH care, in the event of a pandemic situation. It is critically important to have clear policy guidelines to protect and motivate CHWs, the frontline warriors, to support better performance. Issues of work overload could be addressed through a proper human-resource management policy. CHWs can be better protected by ensuring the availability of adequate PPE kits through strengthened supply-chain management system and policies covering CHW health insurance. Transportation facilities for CHWs, and regular training and updates on the pandemic with real facts would help CHWs handle the situation better. Moreover, socio-behavioral change communication (SBCC) through community involvement will not only improve healthcare utilization but also encourage workers and give them a moral boost to ensure uninterrupted, quality MCH service delivery, even during pandemics.

## 5. Conclusions

The COVID-19 pandemic serves as a lesson for us to strengthen our health system and be prepared for uninterrupted routine health service delivery and utilization, even during any such health emergency situation in the future. Development and implementation of context-appropriate strategic guidelines to address the challenges of frontline warriors will improve their work performance. Appropriate administrative measures through interdepartmental coordination would be effective for better community engagement in order to achieve uninterrupted health services, including MCH care, even in the event of any such health emergency.

## Figures and Tables

**Figure 1 healthcare-10-00088-f001:**
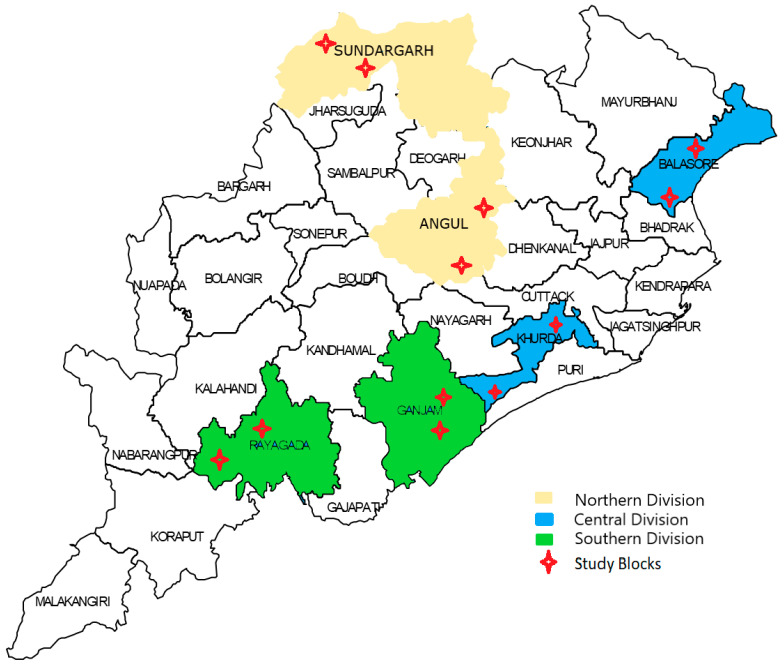
Map of Odisha and the division-wise study districts and blocks.

**Figure 2 healthcare-10-00088-f002:**
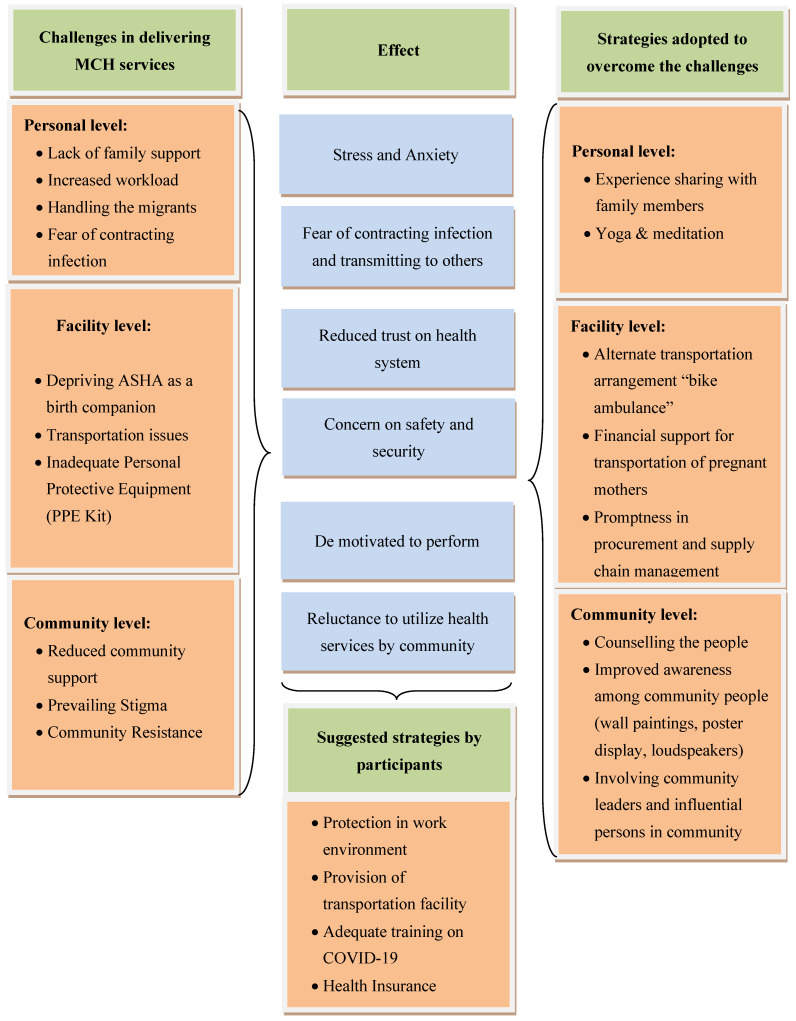
Conceptual framework depicting the challenges faced, their effect, and strategies adopted by CHWs.

**Table 1 healthcare-10-00088-t001:** Detailed descriptions of study sites and participants.

Revenue Division	Name of Study District	Name of Study Block	Data Collection Method (*n*)
Northern	Sundargarh	Bargaon	IDI—LHV (*n* = 1) FGD—ASHA (*n* = 8)
		Balisankara	IDI—LHV (*n* = 1) FGD—ANM (*n* = 7)
	Angul	Angul	IDI—LHV (*n* = 1) FGD—ASHA (*n* = 9)
		Talcher	IDI—LHV (*n* = 1) FGD—ANM (*n* = 8)
Central	Khordha	KhordaSadar	IDI—LHV (*n* = 1) FGD—ASHA (*n* = 8)
		Chilika	IDI—LHV (*n* = 1) FGD—ANM (*n* = 10)
	Balasore	Remuna	IDI—LHV (*n* = 1) FGD—ANM (*n* = 8)
		Simulia	IDI—LHV (*n* = 1) FGD—ASHA (*n* = 9)
Southern	Ganjam	Hinjilikatu	IDI—LHV (*n* = 1) FGD—ANM (*n* = 7)
		Purusottampur	IDI—LHV (*n* = 1) FGD—ASHA (*n* = 8)
	Rayagada	Kalyansinghpur	IDI—LHV (*n* = 1) FGD—ASHA (*n* = 10)
		Kashipur	IDI—LHV (*n* = 1) FGD—ANM (*n* = 8)

## Data Availability

The data presented in this study are available on request.

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
