# Peer review of "Resolution of Resilience: Empirical Findings on the Challenges Faced and the Mitigation Strategies Adopted by Community Health Workers (CHWs) to Provide Maternal and Child Health (MCH) Services during the COVID-19 Pandemic in the Context of Odisha, India"

_healthcare, 2022, doi:10.3390/healthcare10010088_

Round 1

Reviewer 1 Report

This is an interesting study, using sound methodology and presenting well the results of a qualitative research regarding  the challenges
faced and the mitigation strategies adopted by the Community
Health Workers  to provide maternal and child health
 services during COVID-19 pandemic in India.

The results have clear implications for research and practice.

The introduction should offer briefly some information about the routine activities of the community workers and the differences between the three types of medical centers where the study was performed.

Author Response

Point 1: This is an interesting study, using sound methodology and presenting well the results of a qualitative research regarding the challenges faced and the mitigation strategies adopted by the Community Health Workers to provide maternal and child health services during COVID-19 pandemic in India.

 Response 1: Thank you for reviewing our manuscript and providing valuable inputs. Thank you so much for appreciating.

Point 2: The results have clear implications for research and practice.

Response 2: Thank you so much for your kind words.

Point 3: The introduction should offer briefly some information about the routine activities of the community workers and the differences between the three types of medical centers where the study was performed.

Response 3: This suggestion is addressed in the revised manuscript.

Please refer- Introduction, page-3, Line 48-54 and Methodology, Page-4, Line 88-90.

Reviewer 2 Report

I think authors need to bit improve this paper. 

I think this is very interesting topic "Resolution of resilience: Empirical findings on the challenges faced and the mitigation strategies adopted by the Community Health Workers (CHWs) to provide maternal and child health (MCH) services during COVID-19 pandemic in the context of Odisha, India". I have mention some points on thus manuscript.

  1. First of all in the introduction part, Author(s) need to improve the COVID-19 importance. Please must tell about the relationships in this part. According to the paper, this is not enough and please rewrite the some parts of the introduction.
  2. Data collection procedures are not enough, please write in proper way. What is the procedures of data collection .
  3. Challenges, I am pleased for this part. I think it's enough challenges for present study.
  4. Implications are not enough for this study. Please write more about the implications of this study.

Author Response

Point 1: I think this is very interesting topic "Resolution of resilience: Empirical findings on the challenges faced and the mitigation strategies adopted by the Community Health Workers (CHWs) to provide maternal and child health (MCH) services during COVID-19 pandemic in the context of Odisha, India". I have mentioned some points on thus manuscript.

 Response 1: Thank you for reviewing our manuscript and providing your valuable inputs. Thanks for appreciating.

Point 2: First of all in the introduction part, Author(s) need to improve the COVID-19 importance. Please must tell about the relationships in this part. According to the paper, these are not enough and please rewrite the some parts of the introduction.

Response 2: As per your suggestion, we have included the points in the revised version.

Please refer to Introduction, page-3, Line 61-65.

Point 3: Data collection procedures are not enough, please write in proper way. What is the procedure of data collection?

Response 3: We have addressed this in the revised one under the data collection section.

Please refer Data collection under Methodology, page-6&7, Line- 97-109.

Point 4: Challenges, I am pleased for this part. I think it's enough challenges for present study.

Response 4: Thank you very much. Further revision is done based on comment from other reviewer.

Point 5: Implications are not enough for this study. Please write more about the implications of this study.

Response 5: Thank you for the suggestion. In the revised paper, we are renaming the implications as effects to avoid any confusion, as suggested by other reviewer. Points under this section are now detailed and elaborated.

Please refer Discussion section, Page-14 &15, Line-264-276

Reviewer 3 Report

Strengthening the ability building of front-line workers is an indispensable part of combating the COVID-19 epidemic. It is a pleasure to know some details of the current situation in India through this paper although it is just the tip of the iceberg. Regretfully, this manuscript, in my view, still has some major defects. My specific comments are below.

  1. The service capacity building of community health workers(CHWs) should be paid attention to, and it is also necessary to make clear the current challenges faced by CHWs. However, there are still significant limitations to what the authors did in this study. I think this manuscript reads like a work report rather than an academic paper for lack of needful theoretical exploration. In other words, this study has neither an analysis framework nor the necessary theoretical support.
  2. The structure arrangement is also questionable, which leads to the confusing composition and content of this paper. In this paper, there is no clear logic between different parts, especially between 4-7 parts (“4.Challenges”, “5. Implications”, “6. Strategies”, and “7.Implications for policy and practices” ). The structure should be redesigned, and the logicality should be improved to a large extent.

         Furthermore, the paper has some features which do not accord with the           standard. The readability of the paper also needs to be improved. 

Author Response

Point 1: Strengthening the ability building of front-line workers is an indispensable part of combating the COVID-19 epidemic. It is a pleasure to know some details of the current situation in India through this paper although it is just the tip of the iceberg. Regretfully, this manuscript, in my view, still has some major defects. My specific comments are below.

Response 1: Thank you for reviewing our manuscript and providing valuable inputs. We have tried our best to revise it, to make it more informative and useful for the scientific community. We hope the revised one satisfies your expectation.

Point 2: The service capacity building of community health workers(CHWs) should be paid attention to, and it is also necessary to make clear the current challenges faced by CHWs.

However, there are still significant limitations to what the authors did in this study. I think this manuscript reads like a work report rather than an academic paper for lack of needful theoretical exploration. In other words, this study has neither an analysis framework nor the necessary theoretical support.

Response 2: Thank you. Need for capacity building (training and factual update) for CHWs has clearly emerged in our study finding and is highlighted.

Please refer- Results, Page-12, Line- 225-226, Discussion, Page 16, Line- 305 & 316.

To bring out the current challenges encountered by CHWs in the event of COVID-19 pandemic was the objective of this study. Revision is done to clearly depict the challenges.

Please refer- Methodology, page-14, Line- 252-262.

Specifying the limitations would have helped us to improvise the manuscript. However, we have revised it with best efforts to make it better.

As per your comment, the manuscript is revised to attain a better theoretical exploration. Based on the challenges faced by the CHWs, its effect and strategies adopted, a conceptual framework has been developed and discussed in the revised paper.

Please refer to Figure-2, page-13.

Point 3: The structure arrangement is also questionable, which leads to the confusing composition and content of this paper. In this paper, there is no clear logic between different parts, especially between 4-7 parts (“4.Challenges”, “5. Implications”, “6. Strategies”, and “7. Implications for policy and practices”). The structure should be redesigned, and the logicality should be improved to a large extent.

Response 3: As per the suggestion, the structure is now redesigned and properly arranged according the journal requirement. The numbering for the 4-7 is removed and content is systematically presented to improve the readability. Implication is now renamed as Effects to avoid any confusion.

Please refer to page- 14-16 under methodology section.

Point 4: Furthermore, the paper has some features which do not accord with the standard. The readability of the paper also needs to be improved.

Response 4: We have put our best effort to revise the paper for better readability. Specific comments would have helped us even more.

Reviewer 4 Report

Dear Authors,

your article is very interesting and inspiring. Some points were noticed during the review, which I list below and suggest their revision.

Theoretical background

Background on the health care situation and especially the role and tasks of the CHW should be explained; also the selected region should be characterized: What are the specifics both in terms of maternal and infant health and in terms of the health care situation.

The research gap should be concisely elaborated and a concise research question for the qualitative study should be included.

Methodology

Please ensure that any abbreviations used (e.g., ASHA, ANM, LHV) are explained. What are the responsibilities of these health professionals? Details on the recruitment of interview participants are missing; here it should be addressed - as far as possible for data protection reasons - which interview partners were preferred and how this was realized.

The number of participants (interviews and focus group) should be added to Table 1.

The interview guide should be briefly outlined.

"The researchers transcribed the audio recordings in Odia and later translated those into English." How was it ensured that no bias in the statements of content occurred as a result of the translation.

The selected method of qualitative content analysis should be mentioned; in addition, the conducted analysis should also be briefly described.

Results

It would be interesting to know whether the results differ between the interviews and the focus groups and whether there are differences between the different CHW.

Figure 1: Presentation should be corrected; texts are cut off.

Challenges

This section is very brief; it would be interesting to see a comparison between the challenges and strategies chosen, while taking into account the context. Especially in the aspect of stigma, it would be highly interesting to know what strategies the women have chosen to deal with it.

Implication

"The challenges may affect the moral boost and motivation of CHWs in performing their assigned work for the community. What statement should be made here." Some elaboration of this should be done and be put in the context of the results.

"A study conducted by Gupta et al [13] highlighted some of the common mental health problems such as anxiety, depression, stress-related disorders among CHWs." Is there any evidence of mental health problems among CHWs in the interviews.

Strategies

This section is confusing because the direct reference to the findings is missing - or unclear. It appears that some of the measures were already in place. Were they already available to the CHW at the time of the survey? This section should be revised in this respect.

Methodological consideration

"We followed the source and investigators triangulation approach to enhance the credibility of study findings." The approach should be included and briefly described in the methods section. A critical reflection of the study is missing and should be added.

Author Response

Point 1: Dear Authors, your article is very interesting and inspiring. Some points were noticed during the review, which I list below and suggest their revision.

 Response 1: Thank you for reviewing our manuscript and providing us your valuable detailed inputs. Thanks for appreciating.

Point 2: Theoretical background

Background on the health care situation and especially the role and tasks of the CHW should be explained; also the selected region should be characterized: What are the specifics both in terms of maternal and infant health and in terms of the health care situation.

Response 2: Thank you. As suggested, we have included the facts pertaining to MCH care, the roles and responsibilities of different cadres of CHWs during COVID-19 pandemic.

Please refer to Introduction, Page-3, Line 61-65.

The information on the study regions has also been included in figure-1 and table-1. However information in terms of maternal and infant health care in study areas has been given in general context not specific to study area.

Please refer to Methodology, Page-4, Line-61-65 and Page- 5, Figure-1, Table-1.

Point 3: The research gap should be concisely elaborated and a concise research question for the qualitative study should be included.

Response 3: Addressed in the revised one.

Please refer to Introduction, Page-3, Line- 74-77.

Point 4: Methodology

Please ensure that any abbreviations used (e.g., ASHA, ANM, LHV) are explained. What are the responsibilities of these health professionals? Details on the recruitment of interview participants are missing; here it should be addressed - as far as possible for data protection reasons - which interview partners were preferred and how this was realized.

Response 4: Abbreviations and roles of the health professionals are now explained. Detailed numbers and types of the interview participants (site wise) and methods of data collection are given under the methodology sections and also detailed in Table 1. The process of recruitment of study participants is now explained. Thank you.

Please refer to Introduction, Page-3, Line- 48-55.

Point 5: The number of participants (interviews and focus group) should be added to Table 1.

Response 5: This is now addressed.

Please refer to Table-1, page- 5 & 6.

Point 6: The interview guide should be briefly outlined.

Response 6: This is now addressed.

Please refer to Methodology, Page-6, Line- 98-101.

Point 7: "The researchers transcribed the audio recordings in Odia and later translated those into English." How was it ensured that no bias in the statements of content occurred as a result of the translation?

Response 7: Thank you. The researchers with sound knowledge in both Odia and English language transcribed and translated the data. The translations were cross verified by the project investigators. We followed the source and investigators triangulation approach to enhance the credibility of study findings. Detailed in the methodology section.

Please refer to Methodology, Page-7, Line- 109-113.

Point 8: The selected method of qualitative content analysis should be mentioned; in addition, the conducted analysis should also be briefly described.

Response 8: A qualitative content analysis method following conventional approach was adopted to analyze the collected data. As per your suggestion, this section is now revised.

Please refer to Methodology, Page-7, Line- 113,118.

Point 9: Results

It would be interesting to know whether the results differ between the interviews and the focus groups and whether there are differences between the different CHW.

Response 9: Yes, though there were differences of opinion in response to some questions, as can be seen in the results section. For example, some participants expressed their fear of the infection, whereas some others didn’t have much fear, and they took necessary precautions and carried out their tasks with strong will power. We have included your suggestion in the revised one. Thank you.

Please refer to Results, Page-8, Line- 130-133.

Point 10: Figure 1: Presentation should be corrected; texts are cut off.

Response 10: This is now addressed. The figure number has been revised under Figure-2. Hope the texts are now visible.

Please refer to figure-2, Page-13

Point 11: Challenges

This section is very brief; it would be interesting to see a comparison between the challenges and strategies chosen, while taking into account the context. Especially in the aspect of stigma, it would be highly interesting to know what strategies the women have chosen to deal with it.

Response 11: Thank you for your suggestion. This section is now revised as per the suggestion. Hopefully it now addresses your query.

Please refer to Discussion, Page-14, Line- 251-261.

Point 12: Implication

"The challenges may affect the moral boost and motivation of CHWs in performing their assigned work for the community. What statement should be made here?" Some elaboration of this should be done and be put in the context of the results.

Response 12: We have revised this section under the subheading Effects instead of implications as suggested by other reviewer. Hopefully the revised one addresses your point.

Please refer to- Discussion, page- 14 & 15, Line- 263-275.

Point 13: "A study conducted by Gupta et al [13] highlighted some of the common mental health problems such as anxiety, depression, stress-related disorders among CHWs." Is there any evidence of mental health problems among CHWs in the interviews?

Response 13: The study findings like fear of contracting the infection, stress and anxiety clearly suggest about prevailing mental health problems among CHWs. This is now addressed in the revised paper.

Please refer to- Discussion, page-14, Line- 263-268.

Point 14: Strategies

This section is confusing because the direct reference to the findings is missing - or unclear. It appears that some of the measures were already in place. Were they already available to the CHW at the time of the survey? This section should be revised in this respect.

Response 14: This section is now revised as per your suggestion. Hope the things are clear now.

Please refer to- Discussion, page-14 & 15, Line- 277-302.

The strategies suggested by the participants were not existing or implemented before.

Point 15: Methodological consideration

"We followed the source and investigators triangulation approach to enhance the credibility of study findings." The approach should be included and briefly described in the methods section. A critical reflection of the study is missing and should be added.

Response 15: The suggested approach is now part in methodology section.

Please refer to methodology, page-7, Line-112-114.

Though we haven’t put a separate section on the critical reflections, the revised write-up in results and discussion section clearly illustrates this. Thank you.
